# Optimization of Spray-Drying Process with Response Surface Methodology (RSM) for Preparing High Quality Graphene Oxide Slurry

**Xingxing Ye, Yexun Shi, Liming Shen \*, Peng Su and Ningzhong Bao \***

State Key Laboratory of Material-Oriented Chemical Engineering, College of Chemical Engineering, Nanjing Tech University, Nanjing 210009, China; 201861104053@njtech.edu.cn (X.Y.); shiyexun@njtech.edu.cn (Y.S.); 201961204182@njtech.edu.cn (P.S.)
**\*** Correspondence: lshen@njtech.edu.cn (L.S.); nzhbao@njtech.edu.cn (N.B.); Tel.: +86-25-8317-2244 (N.B.)

**Abstract:** The "Drying-redissolution" method is promising for the industrial production of high-concentration well-dispersed graphene oxide slurry (GOS). As the potential key step in this method, the spray drying process requires a statistical investigation which guides the large-scale preparation of graphene oxide powder (GOP). This work systematically studies the effects of operating parameters, including nozzle airflow rate (439–895 L·h$^{-1}$), atomization pressure (0.5–0.7 MPa), and liquid feed rate (3.0–9.0 mL·min$^{-1}$), by using the response surface methodology integrated Box–Behnken design (RSM–BBD), aiming to produce GOP with high yield and easy re-dispersion. The optimized spray drying condition is predicted to be 439 L·h$^{-1}$, 0.59 MPa, and 9.0 mL·min$^{-1}$, at which a powder yield of 70.45% can be achieved. The experimentally obtained GOP has an average particle size of 11.65 μm and the low crumpling degree of the particle morphology results in the good re-dispersibility (97.95%) and excellent adsorption performance (244.1 mg·g$^{-1}$) of GOP. The GOS prepared by the spray-dried GOP possess low viscosity and high exfoliation efficiency with a single-layer fraction up to 90.8%, exhibiting good prospects for application. This work first applied the RSM–BBD model on the spray drying process of GO, and evidenced the possibility of producing high-quality GO slurry with the "drying-redissolution" method.

**Keywords:** graphene oxide; spray drying; response surface methodology; drying and redissolution

## 1. Introduction

Graphene oxide (GO) is an important precursor for the preparation of graphene by the redox method, and it is also called functionalized graphene due to the existence of hydroxyl and carboxyl groups covalently bonding to the planar carbon network [1].These abundant oxygen functional groups are not only beneficial to subsequent modification and dispersion processes but also serve as new active sites, endowing GO broad application prospects in energy storage and conversion, composites, adsorption, etc. [2–8]. In order to realize wide applications of GO, a high-concentration well-dispersed GO dispersion is often required for forming macroscopic assembly.

Well-dispersed GO dispersion obtained directly from a redox process usually exists in a relatively low mass concentration. Our group successfully concentrated GO dispersion to 16 g·L$^{-1}$ GO slurry (GOS) through cross-flow membrane filtration [9]. However, the sharp increase of the viscosity during the water removal process is detrimental for the further concentration of GOS, thereby resulting in an upper limit of concentration of 20 g·L$^{-1}$ [9]. Instead of concentrating GO dispersion to GOS, the "drying and redissolution" method seems more promising in preparing high-concentration GOS by dispersing GO powder (GOP) in water [10–12]. In this approach, the GO dispersion is first converted into GOP via a drying process and then the obtained powder redisperses in water to prepare GOS with all desired concentrations for practical applications (Figure 1). However, due to the

inevitable stacking issue of GO sheets during the drying process, developing a powder preparation technology that allows easy re-dispersion and exfoliation of GO sheets is critical for the wide application of the "drying and redissolution" method [13,14].

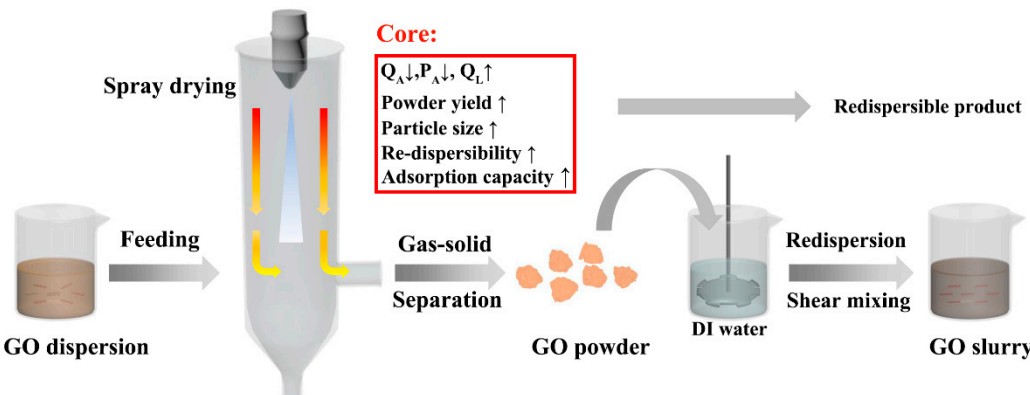

**Figure 1.** Graphic illustration of the "drying and redissolution" method for the production of high-quality GO slurry.

As an efficient powder preparation technology, the spray drying process has been systematically studied regarding the stacking assembly of GO sheets [15]. Shi et al. found that the GO sheets assembled in a relatively ordered manner could ensure the re-dispersibility of GOP by adjusting the feed concentration (>0.8 g·L$^{-1}$), which shed light on the preparation of high quality GOS with the "drying and redissolution" method [16]. However, the evaluation of a powder preparation technology for industrial applications demands more than just the morphology (crumpling degree) and re-dispersibility of the GOP, and the powder yield and other physicochemical properties of GOP also need to be carefully considered. Therefore, a thorough investigation of the spray drying process, to comprehensively evaluate the impact of operating conditions on the GOP, would set the foundation for the industrial production and applications of high quality GOS.

Response surface methodology (RSM), as a statistically based optimization strategy, is widely used to evaluate the influence of several independent variables on one or more responses, which significantly reduces the trial numbers for analyzing multiple variables and their interactions and is very useful for experimental and industrial designs [17–19]. In this work, single-factor experiments are first performed to determine the appropriate range of operating parameters, including the nozzle flow rate (QA), atomization pressure (PA), and liquid feed rate (QL) which are chosen as independent variables. Then, the RSM integrated Box–Behnken design (RSM–BBD) is applied to assess the cross-influences of the parameters on the powder yield, particle size, re-dispersibility, and adsorption capacity. The drying temperature and feed concentration are fixed at 200 °C and 4 g·L$^{-1}$, respectively (according to our previous studies), in order to ensure the drying efficiency and relatively oriented structure of the GOP [16]. The RSM–BBD model estimate the GOP with a 70.45% powder yield, 11.65 μm particle size, 97.95% re-dispersibility, and 244.1 mg·g$^{-1}$ adsorption capacity when the nozzle airflow rate is 439 L·h$^{-1}$, atomization pressure is 0.59 MPa, and the liquid feed rate is 9.0 mL·min$^{-1}$. The predicted values were successfully confirmed by the experimental results, showing the accuracy of the model. The morphology and chemical properties of spray dried GOP were compared with that of the pristine GO and no significant difference was observed. The GOS prepared by high shear mixing spray-dried GOP with water has the viscosity reduced by 50%, the exfoliation time decreased by 75%, and the relative single-layer fraction as high as 90.8% when compared with the GO dispersion obtained from the direct exfoliation of graphite oxide. This work reveals the comprehensive effects of the operating parameters on the spray drying efficiency and can promote the industrial production of GOS for broad applications.

## 2. Materials and Methods

### 2.1. Chemicals and Reagents

Natural flake graphite around 6.5 μm in size (2000 mesh) was purchased from the Aladdin Industrial Corporation. Potassium permanganate ($KMnO_4$, 99.5%, analytical grade), potassium nitrate ($KNO_3$, 99.5%, analytical grade), sulfuric acid ($H_2SO_4$, 98%), hydrogen peroxide aqueous solution ($H_2O_2$, 30%), and ammonium hydroxide ($NH_3 \cdot H_2O$, 25–28%) were bought from Shanghai Lingfeng Chemical Reagents Company. Methylene blue was purchased from Tianjin Chemical Reagent Research Institute Co., Ltd (Tianjin, China). All chemicals were used as received without further purification.

### 2.2. Experimental Design

Single-factor experiments were performed first in order to determine the appropriate range of the operating parameters including the nozzle airflow rate ($X_1$, $L \cdot h^{-1}$), atomization pressure ($X_2$, MPa), and liquid feed rate ($X_3$, $mL \cdot min^{-1}$). RSM design experiments were carried out to evaluate the influence of the variables on the following four aspects, powder yield ($Y_1$, %), particle size ($Y_2$, μm), re-dispersibility ($Y_3$, %), and adsorption capacity ($Y_4$, $mg \cdot g^{-1}$). A three-level-three-factor Box–Behnken design with 17 runs (5 repetitive runs and 12 factorial runs) was carried by using Design-Expert software (version 12.0.0). The detailed descriptions of the independent factors with coded and actual levels are presented in Table 1.

**Table 1.** Coded and actual levels of independent variables for Box–Behnken design.

| Independent Variables | Levels of Coded Variables ($x_i$) | | |
|---|---|---|---|
| | **−1** | **0** | **1** |
| $X_1$: nozzle airflow rate/$L \cdot h^{-1}$ | 439 | 667 | 895 |
| $X_2$: atomization pressure/MPa | 0.5 | 0.6 | 0.7 |
| $X_3$: liquid feed rate/$mL \cdot min^{-1}$ | 3.0 | 6.0 | 9.0 |

To predict the responses ($Y$), the mathematical quadratic polynomial models were utilized as follows:

$$Y = \beta_0 + \sum_{i=1}^{3} \beta_i x_i + \sum_{i=1}^{3} \beta_{ii} x_i^2 + \sum_{i=1}^{3} \sum_{j=i+1}^{3} \beta_{ij} x_i x_j, \tag{1}$$

where $\beta_0$ is the value of fitted response at the design central point (0,0,0), $\beta_i$, $\beta_{ii}$, and $\beta_{ij}$ are the coefficients of linear effect, quadratic effect, and interactive effect, respectively, and $x_i$ is the dimensionless coded value of the independent variable. The test of significance was conducted on total error criteria at confidence level 95% [20,21]. Analysis of variance (ANOVA) was applied to evaluate the significance of the obtained model.

### 2.3. Sample Preparation and Spray Drying

The graphite oxide was prepared with the modified Hummers method and then washed with DI water until it became neutral [22,23]. The sample was diluted in DI water to form a suspension with a mass concentration of 4 $g \cdot L^{-1}$, and then ultrasonicated for 2 h in an ultrasonicated bath (KQ-300B, Kunshan, China) to obtain GO dispersion. The spray drying of the GO dispersion was performed using a spray-dryer BÜCHI B-290 (Flawil, Switzerland). For each experiment, the feeding volume was fixed at 500 mL, the inlet temperature was set at 200 °C, and the aspiration flow rate was maintained at 40 $m^3 \cdot h^{-1}$. To investigate the effects of the operating parameters, the nozzle airflow rate, atomization pressure, and liquid feed rate were adjusted in the range of 283–1051 $L \cdot h^{-1}$, 0.4–0.8 MPa, and 3.0–9.0 $mL \cdot min^{-1}$, respectively. Before drying the GO dispersion, the dryer was run for 10 min by feeding DI water to obtain a steady-state condition. After the drying process, the GOP was collected from the collection vessel and then stored in a humidity chamber

(HWS-080, Jinghong, China) with a relative humidity of 70% and temperature of 25 °C for at least 48 h. To prepare the graphene oxide slurry (GOS), 12.5 g GOP was dispersed in 500 mL DI water, and the pH was adjusted to 8 using ammonium hydroxide. The above mixture was first dispersed at 500 rpm for 5 min, then sheared at 3000 rpm for 60 min, and the obtained GOS was denoted by GOS-1. For comparison, another GOS was prepared by directly shearing graphite oxide dispersion (25 g·L$^{-1}$) at 3000 rpm for 60 min, and the obtained sample was denoted by GOS-2.

*2.4. Characterizations*

The analysis of GOP such as powder yield, particle size, re-dispersibility, and adsorption capacity were shown detailed in Discussion S1 in the supporting information section. The morphology and structure of the GOP samples were observed with a field-emission scanning electron microscope (FESEM, HITACHI S-4800). Thermo-gravimetric analysis (TGA) was performed using an STA 449C analyzer (NETZSCH, Selb, Germany) from 25 to 800 °C at a heating rate of 10 °C·min$^{-1}$ in the nitrogen atmosphere. X-ray photoelectron spectroscopy (XPS) analysis was carried on a PHI 5000 Versa Probe III X-ray photoelectron spectrometer (ULVAC–PHI Inc., Kanagawa, Japan). For the moisture content test, the obtained GOPs after drying were directly placed in an oven (60 °C, 4 h), and the weight differences before and after the heat treatment were recorded. Viscosity measurements were carried out by using a DV2T Brookfield laboratory viscometer with a CAP-52Z cone spindle (shear rate: 2.00 N s$^{-1}$, sample volume: 0.5 mL, cone angle: 3°, and cone radius: 1.2 cm). The relative single-layer fraction was calculated by the concentration ratio of GO supernatant and GO slurry. UV tests were performed on the Cary300 (Agilent, Santa Clara, CA, USA) UV–Visible spectrophotometer.

## 3. Results and Discussions

*3.1. Single-Factor Experiments*

The property of the GOP obtained from the spray drying process is mainly affected by operating parameters such as nozzle airflow rate ($X_1$), atomization pressure ($X_2$), and liquid feed rate ($X_3$) [24]. The influences of these operating parameters on the powder yield ($Y_1$), particle size ($Y_2$), re-dispersibility ($Y_3$), and adsorption capacity ($Y_4$) are presented in Figure 2. As shown in Figure 2a, $Y_1$ first increases from 41.63 to 68.80% when $X_1$ is increased from 283 to 439 L·h$^{-1}$ and then progressively drops to 43.27% with the continuous increase of $X_1$. The low yield at 283 L·h$^{-1}$ is caused by the corresponding droplet size (Table S1) because a large droplet will stick to the wall of drying chamber before being dried. The droplet sizes at different spray drying conditions were calculated by using an empirical formula for the bi-fluid nozzle (see details in Discussion S2) [25]. The droplet decreases in size as $X_1$ is increased (Table S1), just as for the resulted particle size. As $X_1$ is increased beyond 439 L·h$^{-1}$, the smaller particle size limits the efficiency of solid-gas separation in the cyclone separator, thus reducing $Y_1$ [26]. Figure 2b displays that the re-dispersibility of GOP ($Y_3$) gradually deteriorates from 92.81 to 4.35% as $X_1$ is increased from 283 to 1051 L·h$^{-1}$. The re-dispersibility is related to the particle size and crumpling degree of GOP [16]. The particle size decreases because the crumpling degree of each GOP increases as $X_1$ is increased (Figure 3a–c). The highly crumpled structure will inhibit the redispersion of GOP, resulting in the exacerbation of the re-dispersibility. An incomplete unfolding of GOP during the redispersion process leads to the decrease in the final adsorption performance of the GOP ($Y_4$). Considering the above four aspects ($Y_1$–$Y_4$), $X_1$ in the range of 439–895 L·h$^{-1}$ is selected in the corresponding experiments of RSM.

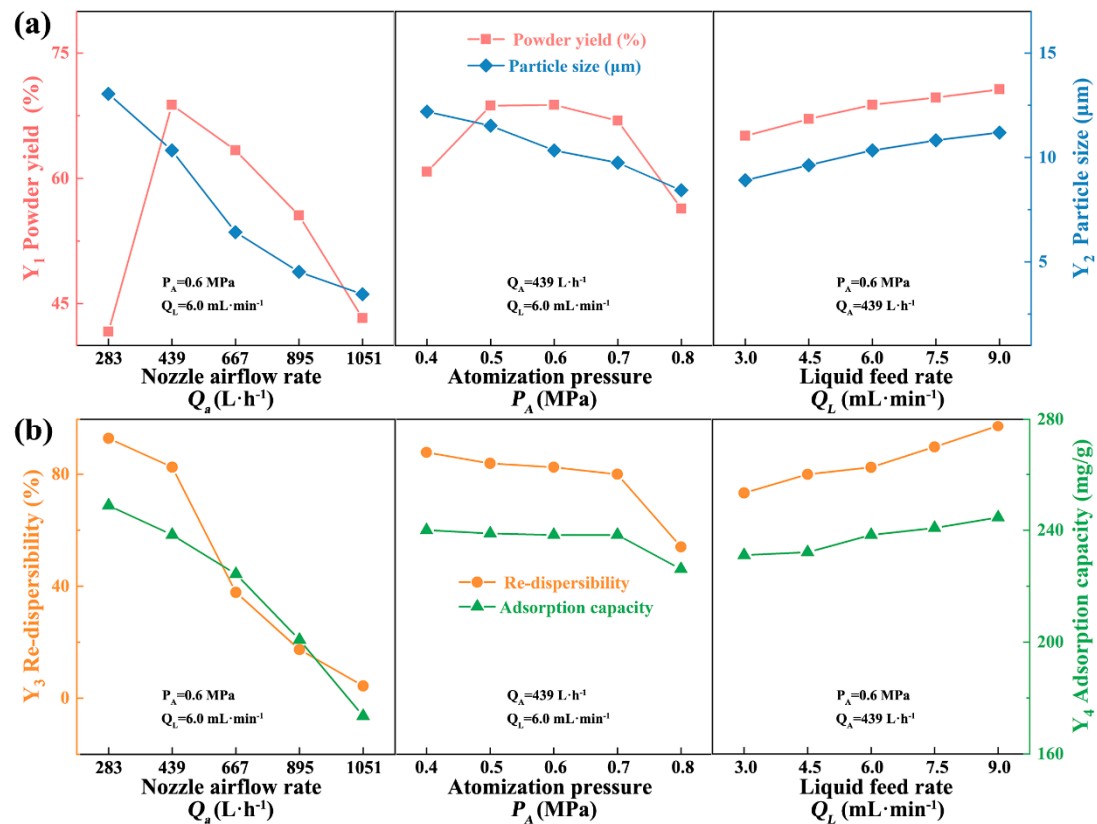

**Figure 2.** The influence of nozzle airflow rate, atomization pressure, and liquid feed rate on the (**a**) powder yield ($Y_1$) and particle size ($Y_2$) and (**b**) re-dispersibility ($Y_3$) and adsorption capacity ($Y_4$).

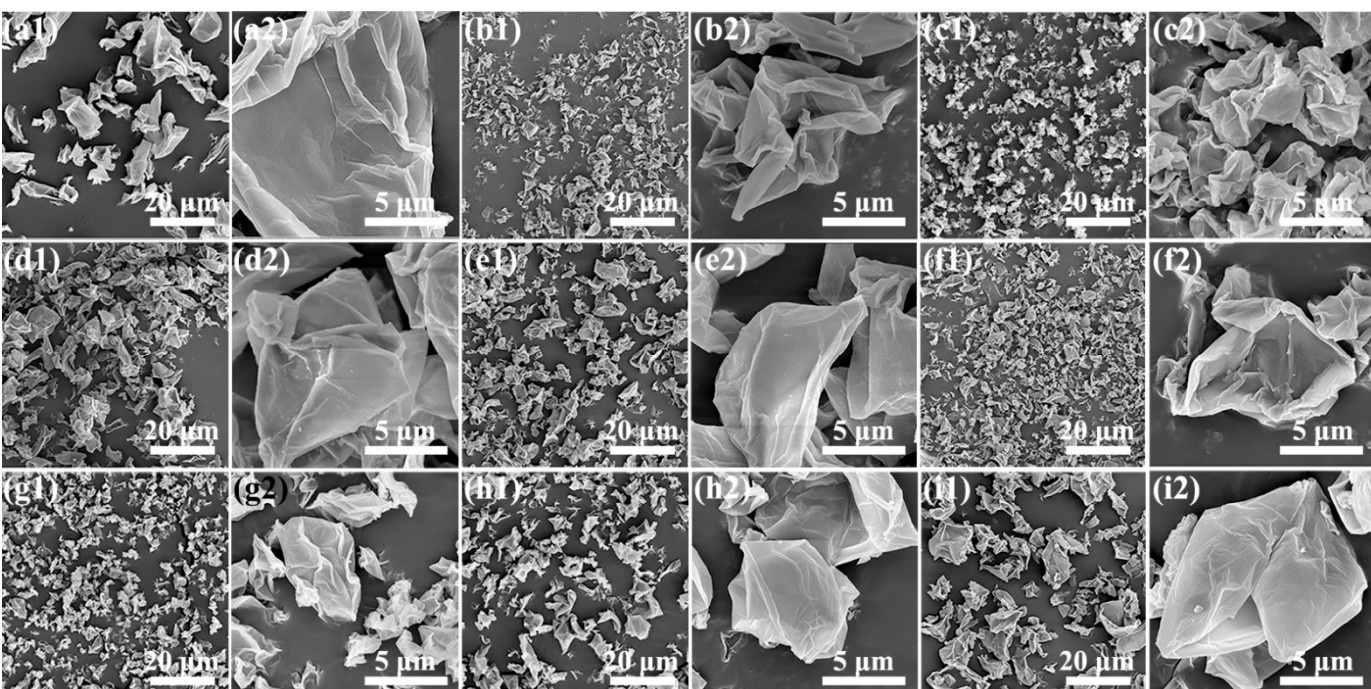

**Figure 3.** SEM images of GOPs with different (**a**–**c**) nozzle airflow rates/$X_1$ (**a1**,**a2**: 283 L·h$^{-1}$; **b1**,**b2**: 667 L·h$^{-1}$; **c1**,**c2**: 1051 L·h$^{-1}$), (**d**–**f**) atomization pressures/$X_2$ (**d1**,**d2**: 0.4 MPa; **e1**,**e2**: 0.6 MPa; **f1**,**f2**: 0.8 MPa), and (**g**–**i**) liquid feed rates/$X_3$ (**g1**,**g2**: 3.0 mL·min$^{-1}$; **h1**,**h2**: 6.0 mL·min$^{-1}$; **i1**,**i2**: 9.0 mL·min$^{-1}$).

The variation of $X_2$ affects the relative velocity between the gas and liquid. Similar to the influence of $X_1$, the increase of $X_2$ also results in the decrease of the droplet size (Table S2). Therefore, $Y_1$ increases first and then decreases, and $Y_2$ drops from 12.20 to 8.43 µm with increase of $X_2$. Since the variation range of the droplet size is narrower than that caused by $X_1$, $Y_2$ does not change significantly. Figure 3d–f show that the crumpling degree of GOPs has slightly increased. Accordingly, $Y_3$ and $Y_4$ are slightly decreased. Therefore, in the subsequent experimental design, the range of $X_2$ is chosen between 0.5 to 0.7 MPa.

The influences of $X_3$ on all dependent variables ($Y_1$–$Y_4$) are positively correlated. With the increase of $X_3$ from 3.0 to 9.0 mL·min$^{-1}$, the droplet size increases from 16.77 to 27.49 µm (Table S3). The large droplets contain more GO nanosheets and subsequently form larger GOP with low crumpling degrees. As shown in the SEM images in Figure 3g–i, the particle size of the GOP is getting larger with the increase of $X_3$. Meanwhile, the increase in droplet size is accompanied by the increase of the dry basis moisture content of GOP from 0.0742 to 0.1159 kg·kg$^{-1}$ (Figure S2), because more energy is required to remove water in the first drying stage when more liquid is fed into the drying chamber, which lowers the temperature for the second drying stage and eventually results in the higher moisture content of GOP [26]. The humidity in GOP improves its wettability, which is conducive to the final redispersion and, at the same time, beneficial to the final adsorption ability as well [20]. The $X_3$ used in the following experiment design is in the range from 3.0 to 9.0 mL.

### 3.2. RSM Analysis
#### 3.2.1. Model Fitting

According to the results of the single-factor experiments, the RSM with BBD was applied to optimize the influences of operating parameters on the powder yield ($Y_1$), particle size ($Y_2$), re-dispersibility ($Y_3$), and adsorption capacity ($Y_4$), and the experimental design and the response results are listed in Table 2. After fitting with the second-order polynomial equation (Equation (1)), empirical models in terms of the coded factors were attained as follows:

$$Y_1 = 63.24 - 6.64x_1 - 1.79x_2 + 3.95x_3 - 0.50x_1x_2 + 1.22x_1x_3 + 1.22x_2x_3 - 1.04x_1^2 - 0.89x_2^2 - 1.11x_3^2 \tag{2}$$

$$Y_2 = 6.66 - 2.85x_1 - 0.50x_2 + 1.15x_3 - 0.21x_1x_2 - 0.10x_1x_3 - 0.16x_2x_3 + 0.89x_1^2 + 0.48x_2^2 - 0.05x_3^2 \tag{3}$$

$$Y_3 = 36.31 - 32.62x_1 - 3.83x_2 + 12.58x_3 - 1.49x_1x_2 - 0.09x_1x_3 - 0.41x_2x_3 + 13.69x_1^2 - 0.32x_2^2 + 2.35x_3^2 \tag{4}$$

$$Y_4 = 223.04 - 18.72x_1 - 4.09x_2 + 9.44x_3 - 1.09x_1x_2 + 2.60x_1x_3 + 1.52x_2x_3 - 3.53x_1^2 - 1.16x_2^2 - 0.86x_3^2 \tag{5}$$

**Table 2.** Experimental results of Box–Behnken design for the response variables.

| Test No. | $x_1$ | $x_2$ | $x_3$ | $Y_1$ (%) | $Y_2$ (µm) | $Y_3$ (%) | $Y_4$ (mg·g$^{-1}$) |
|---|---|---|---|---|---|---|---|
| 1 | 1 | 0 | 1 | 59.53 | 5.90 | 31.21 | 211.4 |
| 2 | 1 | 0 | −1 | 49.09 | 4.00 | 7.52 | 187.5 |
| 3 | 1 | 1 | 0 | 52.91 | 4.13 | 12.52 | 194.1 |
| 4 | 0 | −1 | 1 | 66.08 | 8.95 | 56.21 | 233.1 |
| 5 | 0 | 0 | 0 | 62.52 | 6.57 | 37.77 | 224.3 |
| 6 | 0 | 0 | 0 | 63.66 | 6.18 | 33.32 | 221.0 |
| 7 | 0 | 0 | 0 | 63.54 | 6.92 | 39.46 | 221.2 |
| 8 | 0 | 0 | 0 | 63.12 | 7.20 | 36.52 | 223.6 |
| 9 | 1 | −1 | 0 | 56.72 | 5.63 | 22.35 | 206.1 |
| 10 | 0 | 1 | −1 | 53.96 | 5.55 | 21.30 | 205.9 |
| 11 | 0 | 0 | 0 | 63.36 | 6.42 | 34.50 | 225.1 |
| 12 | 0 | −1 | −1 | 60.75 | 6.13 | 28.96 | 217.1 |
| 13 | 0 | 1 | 1 | 64.19 | 7.74 | 46.91 | 228.0 |

**Table 2.** *Cont.*

| Test No. | $x_1$ | $x_2$ | $x_3$ | $Y_1$ (%) | $Y_2$ (μm) | $Y_3$ (%) | $Y_4$ (mg·g$^{-1}$) |
|---|---|---|---|---|---|---|---|
| 14 | −1 | 0 | −1 | 65.09 | 8.91 | 73.32 | 231.1 |
| 15 | −1 | 0 | 1 | 70.65 | 11.19 | 97.37 | 244.6 |
| 16 | −1 | −1 | 0 | 68.71 | 11.52 | 83.85 | 238.8 |
| 17 | −1 | 1 | 0 | 66.91 | 10.84 | 79.99 | 234.4 |

The significances of these second-order polynomial equations were verified by the F-test from the analysis of variance (ANOVA). As shown in Table 3, all the selected models are extremely significant with $p < 0.0001$. For $Y_1$, all the three coded variables, and their interactions (except for $x_1 \cdot x_2$) and quadratic effects are significant. For $Y_2$ and $Y_3$, all the interactions between three variables and the quadratic effects of $x_2$ and $x_3$ are non-significant. For $Y_4$, the interaction between $x_2$ and $x_3$, and their quadratic effects are all non-significant. Since x1 is the most influential variable, we can determine that among three factors, the nozzle airflow rate ($X_1$) had major effects on all responses.

**Table 3.** ANOVA evaluation of linear, quadratic, and interactive terms for each response variable of the spray drying process.

| Source | df | $Y_1$ | | $Y_2$ | | $Y_3$ | | $Y_4$ | |
|---|---|---|---|---|---|---|---|---|---|
| | | F | *p* | F | *p* | F | *p* | F | *p* |
| Model | 9 | 189.53 | <0.0001 ** | 43.25 | <0.0001 ** | 277.84 | <0.0001 ** | 188.32 | <0.0001 ** |
| $x_1$ [1] | 1 | 1134.48 | <0.0001 ** | 307.25 | <0.0001 ** | 1982.81 | <0.0001 ** | 1261.93 | <0.0001 ** |
| $x_2$ [2] | 1 | 82.13 | <0.0001 ** | 9.32 | 0.0185 * | 27.36 | 0.0012 ** | 60.13 | 0.0001 ** |
| $x_3$ [3] | 1 | 400.61 | <0.0001 ** | 49.92 | 0.0002 ** | 294.73 | <0.0001 ** | 320.56 | <0.0001 ** |
| $x_1 \cdot x_2$ | 1 | 3.25 | - | 0.79 | - | 2.08 | - | 6.50 | 0.0382 * |
| $x_1 \cdot x_3$ | 1 | 19.16 | 0.0032 ** | 0.17 | - | 0.01 | - | 12.16 | 0.0102 * |
| $x_2 \cdot x_3$ | 1 | 19.31 | 0.0032 ** | 0.47 | - | 0.16 | - | 4.19 | - |
| $x_1{}^2$ | 1 | 14.69 | 0.0064 ** | 15.76 | 0.0054 ** | 183.76 | <0.0001 ** | 23.64 | 0.0018 * |
| $x_2{}^2$ | 1 | 10.64 | 0.0138 * | 4.63 | - | 0.10 | - | 2.54 | - |
| $x_3{}^2$ | 1 | 16.65 | 0.0047 ** | 0.05 | - | 5.44 | - | 1.39 | - |
| Lack of fit | | 1.36 | 0.2268 ns | 0.83 | 0.3076 ns | 0.3143 | 0.8156 ns | 1.95 | 0.9035 ns |
| R$^2$ | | 0.996 | | 0.982 | | 0.997 | | 0.996 | |
| C.V. (%) | | 0.90 | | 6.32 | | 4.74 | | 0.68 | |

[1] Nozzle airflow rate, [2] Atomization pressure, [3] Liquid feed rates. ns—Not significant, * significant at $p < 0.05$, ** significant at $p < 0.01$.

To ensure the reliability of these models, lack-of-fit tests were performed and the results are presented in Table 3. All the p-values for the lack-of-fit are >0.05 ($p = 0.2263$, 0.3076, 0.8156, and 0.9035), confirming the significances of all models. Moreover, the precisions of these models are indicated by the coefficient of determination R$^2$, which assess how strong the linear relationship between the experimental and predicted variables. The R$^2$ values of these models are close to 1 (R$^2$ = 0.996, 0.982, 0.997, and 0.996), indicating that the entire response variation can be described by these models. To judge the adequacy of these empirical models, the plots of predicted versus actual values and the normal probability plots of studentized residuals were used. As shown in Figure 4a–d, all correlations between the experimental and predicted values of $Y_1$–$Y_4$ are satisfactory, denoting the preferable fitting between the actual and predicted values. In Figure 4e–h, the residuals follow a normal distribution and the points follow a straight line so that these models are valid [27,28].

### 3.2.2. Response Surfaces and Contour Plots

Powder yield ($Y_1$) directly reflects both the drying and cyclone separation effects at different drying conditions, so it is particularly important. Table 2 shows that $Y_1$ ranges from 49.09 to 70.65% in this study. The ANOVA displays that the linear, quadratic effects of $x_1$, $x_2$, and $x_3$, as well as the interaction effect except for x1 and x2, are significant for $Y_1$ (Table 3). Surface plots in Figure 5(a1–a3) depict the influences of the operating parameters on $Y_1$. It can be easily seen that $x_1$ and $x_2$ have negative effects on $Y_1$ while $x_3$ exhibits a

positive one, which are in good agreement with the single-factor experiment results. The negative effects of $x_1$ and $x_2$ are mainly caused by the small droplet size, which increases the difficulty of the cyclone separation in agreement with Amaro's report [26]. However, the positive impact of $x_3$ is opposite to LeClair's study [29]. The increase of liquid feed rate causes the formation of large droplets, which generally increases the possibility of the wall-sticking of droplets, but the $x_3$ selected in this experimental design can meet the drying requirements well, and the large particles obtained by large droplets will be more conducive to the separation and collection by the cyclone, improving the final powder yield ($Y_1$). Besides, Figure 5(a2) shows that although the interaction of ($x_1$, $x_3$) is positively correlated, Y1 is gradually lowered as $x_1$ and $x_3$ increase. This is mainly because the interaction is not as significant as the linear effects of individual variables, resulting in that $Y_1$ is mainly susceptible to the influence of the dominant linear effect. The response surface variations from Figure 5(a1–a3) indicate that the order of significance is: $x_1 > x_3 > x_2$.

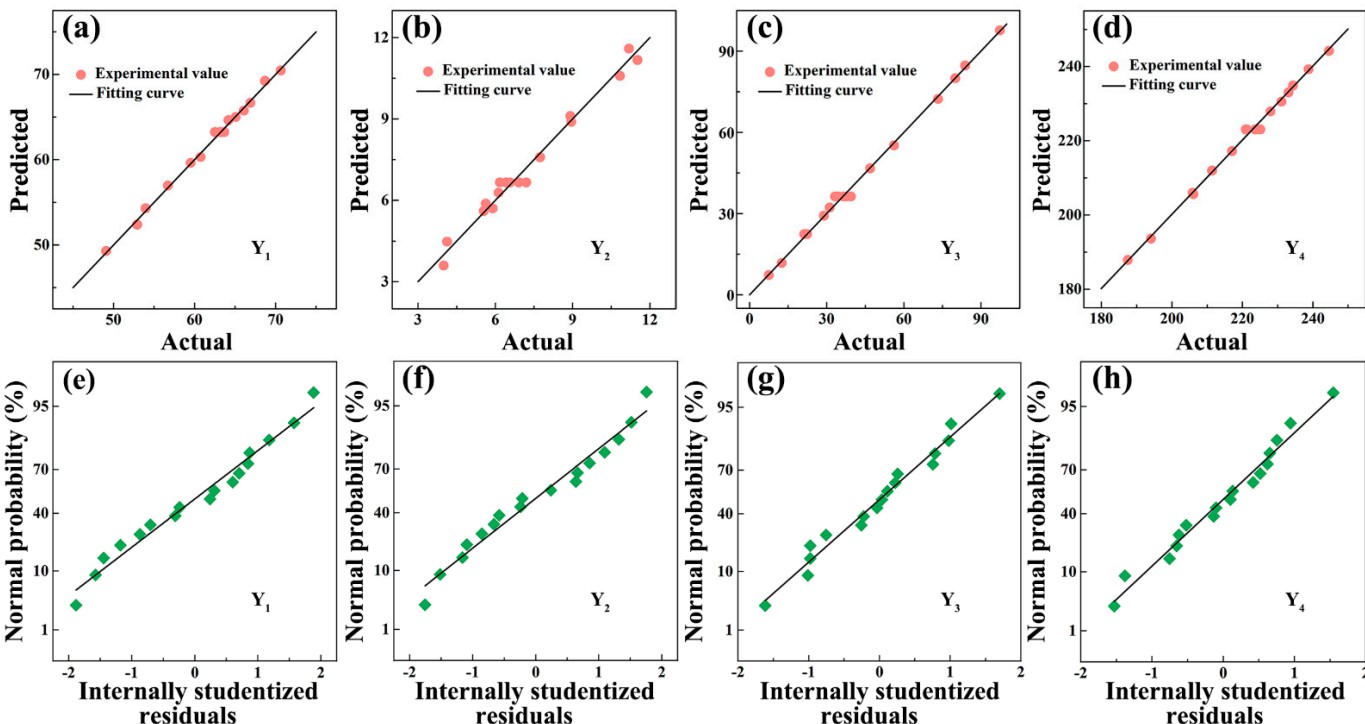

**Figure 4.** (**a**–**d**) Regression graphs of the experimental and predicted values: (**a**) $Y_1$, (**b**) $Y_2$, (**c**) $Y_3$, and (**d**) $Y_4$. (**e**–**h**) Normal probability plots of studentized residuals: (**e**) $Y_1$, (**f**) $Y_2$, (**g**) $Y_3$, and (**h**) $Y_4$.

All the obtained GOPs exhibit sheet-like morphology at the properly fixed feed concentration (4 g·L$^{-1}$). However, the droplet size varies under different atomization conditions, resulting in different particle sizes ($Y_2$) (Figure 3) [23]. The particle size distributions for all the experiments are monomodal with low spans, and $Y_2$ varies from 4.00 to 11.52 μm (Figure S3). ANOVA (Table 3) indicates that only the linear effects of operating parameters are significant at the 95% confidence level, and the order of significance is $x_1 > x_3 > x_2$, similar to the influence on $Y_1$. The interactions between the operating parameters are not significant, indicating no obvious influence on $Y_2$. Therefore, the changes of surface plots in Figure 5(b1–b3) are in good agreement with the results in Figure 2a, which are also consistent with the reports that found the lower atomization level leads to the formation of larger particles [26,30].

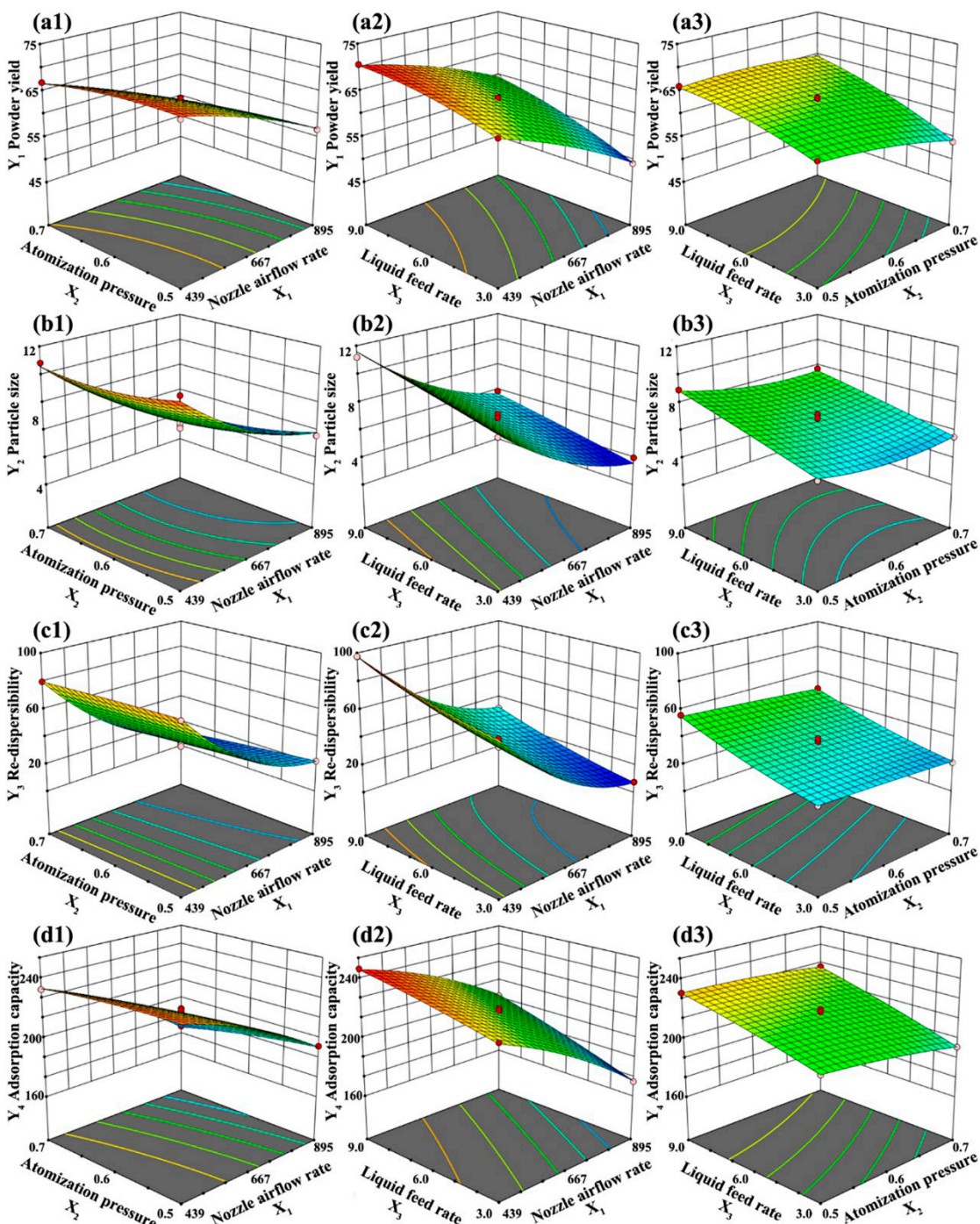

**Figure 5.** Surface plots for statistical models of the influence operating parameters on (**a1**–**a3**) powder yield $Y_1$, (**b1**–**b3**) particle size $Y_2$, (**c1**–**c3**) re-dispersibility $Y_3$, and (**d1**–**d3**) adsorption capacity $Y_4$ against all operating parameters.

The re-dispersibility of the spray-dried GOP is very important for the preparation of high-concentration GOS. It was reported that the re-dispersibility is closely related to its internal microstructure and the crumpling degree, which can be tuned by the droplet size controlled by the operating parameters [16]. ANOVA (Table 3) displays that all the linear effects of variables and only quadratic effects of $x_1$ are significant, and the order of significance is also $x_1 > x_3 > x_2$. At the same time, all the interactions between variables are not significant at the 95% confidence interval, resulting in that the surface plots in Figure 5(c1–c3) exhibit similar changes to Figure 2b. However, it conflicts with Al-Asheh's report, which considers that the increase in droplet size will result in a thicker diffusion

boundary layer and retard the transport of dissolved materials from the particle surface, thus resulting in a decline in solubility [31,32]. In this work, due to the unique layered structure of GOP, the large particle with a low crumpling degree caused by the large droplet size is more favorable for the unfolding process, resulting in a better redispersion.

GO nanosheet is negatively charged, which makes it a material of choice for adsorption due to the electrostatic interactions [33]. It can be seen from ANOVA (Table 3) that besides the influences of significant variables, the interactions of $(x_1, x_2)$ and $(x_1, x_3)$ also affect $Y_4$. The $x_1$ and $x_2$ interact negatively on $Y_4$, while the influence from $x_1$ and $x_3$ interaction is positive. Besides, the interaction effect is far less significant than the linear effect, thus the surface plot (Figure 5(d1–d3)) change is more similar to the single-factor experiment result (Figure 2b). The order of significance is $x_1 > x_3 > x_2$, the same as that for $Y_1$, $Y_2$, and $Y_3$.

### 3.2.3. Optimization of the Operating Parameters

The operating parameters of the spray-dried GOP were successfully optimized by the second-order polynomial models of RSM using Design-Expert software. During the optimization, the independent variables ($X_1$, $X_2$, and $X_3$) were kept within the range, and the responses ($Y_1$, $Y_2$, $Y_3$, and $Y_4$) were set to be maximized. According to the statistical analysis, the optimized condition can be achieved with the desirability of 0.996 when $X_1$ is 439 L·h$^{-1}$, $X_2$ is 0.59 MPa, and $X_3$ is 9.00 mL·min$^{-1}$. The optimized condition is close to the single-factor situation, which is mainly due to the weak interaction between the variables and the limited impact of the interaction effect on all responses. GOP was prepared at the optimized condition (Figure 6(a1,a2)), and there is a good agreement between the predicted values and the actual experimental values about all responses (Table 4), thus validating the rationality of optimization results.

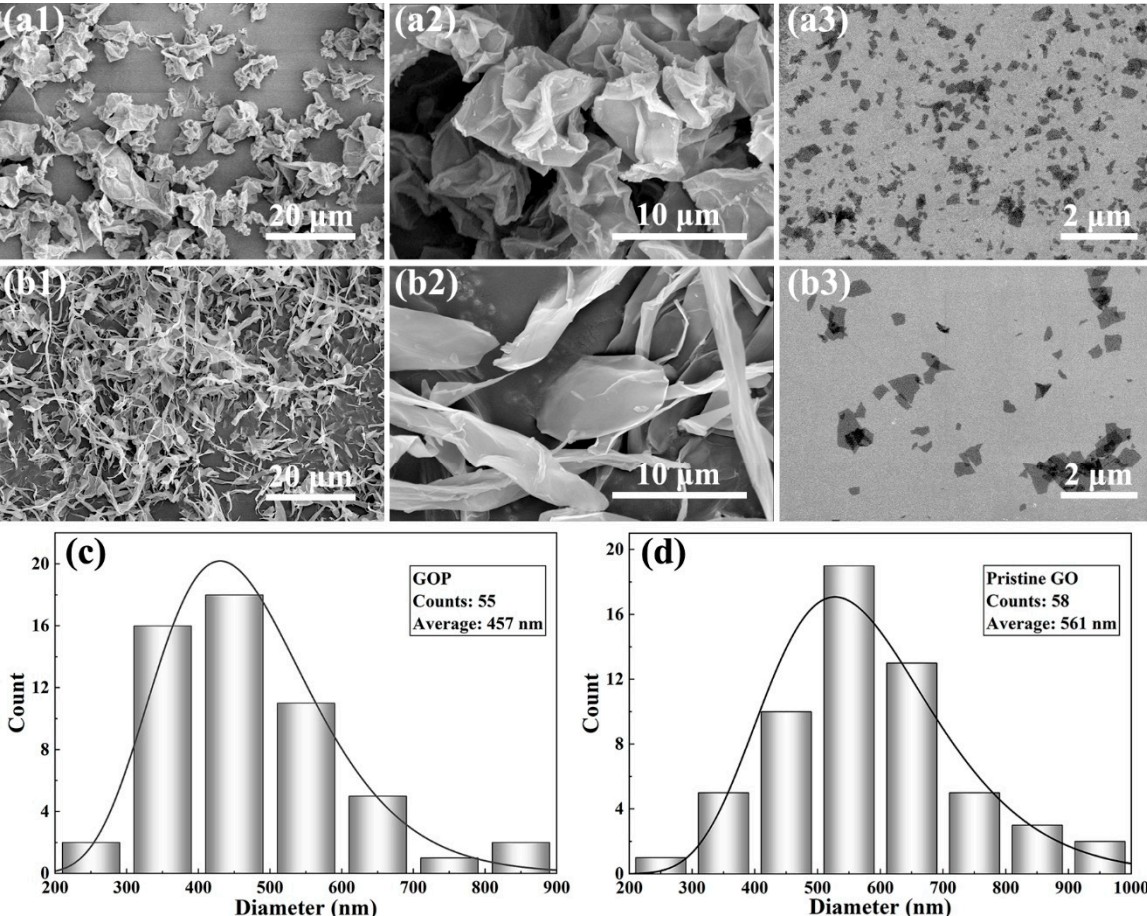

**Figure 6.** The SEM images of the optimized spray-dried GOP and the pristine GO before (**a1**, **a2**, **b1**, **b2**) and after (**a3**, **b3**) redispersion. The lateral size histograms of GOP (**c**) and pristine GO (**d**) after redispersion.

**Table 4.** Comparison between the predicted value and the actual experimental value at the optimized condition.

| Responses | Predicted Value | Actual Value | Deviation |
|---|---|---|---|
| $Y_1$ (%) | 70.45 | 71.96 ± 0.41 | +2.14% |
| $Y_2$ (μm) | 11.65 | 11.06 ± 0.22 | −5.06% |
| $Y_3$ (%) | 97.95 | 96.92 ± 0.79 | −1.05% |
| $Y_4$ (mg/g) | 244.1 | 242.1 ± 0.9 | −0.89% |

*3.3. Analysis of the Optimized Spray-Dried GOP*

3.3.1. Morphology and Chemical Properties of GOP

Figure 6(a1,a2,b1,b2) show the SEM images of the spray-dried GOP and the pristine GO prepared by freeze-drying. By comparing the two samples, it is found that the GOP shows more crumpled morphology, and the particles are smaller and thicker than the pristine GO, indicating that the spray drying is a fast water removal process during which the GO nanosheets are indeed easier to stack and become crumpled. However, both the crumpled GOP and the pristine GO can unfold and exfoliate to GO nanosheets after redispersion in water (Figure 6(a3,b3)), resulting in stable dispersions [34]. The only difference is in their particle sizes. Through size statistics (Figure S4), it is found that the mean size of GOP is 457 nm, which is smaller than the 561 nm of the pristine GO. The size difference is mainly due to the ultrasonic time (2 h for GOP dispersion vs. 30 min for pristine GO dispersion).

To further detect whether the chemical nature of oxygen functionalities had changed after the thermal spray drying, XPS spectra were performed. Figure 7a displays the broad scan XPS spectra of GOP and pristine GO, both samples exhibit two main peaks: C1s and O1s. It can be found that their C/O atomic ratios are the same, indicating a similar chemical composition. High-resolution spectra on the C1s region with curve fittings carried out by using Gaussian–Lorentzian peak shape after the Shirley background correction are shown in Figure 7b [35]. The atomic percentage of each type of carbon presenting in the samples was calculated using area ratios from the C1s region and the data are listed in Figure 7b. For both the GOP and pristine GO, the carbon atoms bound to oxygen in the form of C-O and C=O contribute approximately 44% and 10%, respectively, which indicates that the spraying process did not significantly change the oxygen functionalities of GO and the GOP retains the chemical nature even after the thermal drying process. Figure 7c displayed the Raman spectra of the GOP and pristine GO. It was found that the $I_D/I_G$ ratio of GOP (0.879) was slightly larger than that of the pristine GO (0.834), indicating the existence of more defects and disorder levels in GOP caused by the crumpling morphology [36]. Besides, no obvious 2D peak could be found at ~2700 cm$^{-1}$, which indicated no significant change in oxidation degree of GOP after spray drying. Figure 7d exhibits the TG curves of GOP and the pristine GO. The weight loss can be divided into four parts, corresponding to the weight loss of moisture, hydroxyl/epoxy, organosulfate, and carboxyl [37]. The similar TG curves once again prove that their chemical compositions are similar, indicating no obvious reduction occurred at the optimized drying condition.

3.3.2. Re-Dispersity and Single-Layer Fraction of GOS

The GOS-1 was prepared from GO dispersion by using the "drying and redissolution" method, while GOS-2 was obtained from the direct exfoliation of high-concentration graphite oxide dispersion for comparison. Figure 8a displays the viscosity change with the shear time, and it is found that the viscosities of both samples rise sharply first and then become flat with the extension of shear time. The viscosity of GOS-1 increases more rapidly, reaching 52.7% of the final viscosity (1185 mPa·s) when the shear exfoliation is performed for only 1 min and the corresponding relative single-layer fraction rises to 28.4% (Figure 8b) due to the excellent unfolding and exfoliation ability of the GOP. The 1-min viscosity of GOS-2 is only 6.4% of the final viscosity (2187 mPa·s), and the relative single-layer fraction is 6.7%. After 1 min, the viscosity increase of GOS-1 gradually slows

down, but GOS-2 exhibits a sharp increase and exceeds that of GOS-1 at 6 min. At 6 min, the relative single-layer fraction of the GOS-1 reaches 62.7% while GOS-2 is only 18.1%. At 10 min, the relative single-layer fraction of the GOS-1 reaches 79.9%, corresponding to a GO concentration of about 20 g·L$^{-1}$ which is the upper limit concentration for the direct water removal of the GO dispersion. It takes 40 min for GOS-2 to reach this concentration. By comparison, the shear time of GOS-1 is reduced by 75%. Besides, the relative single-layer fractions of both samples achieve ~90% at 60 min, but the viscosity of GOS-1 is 45.8% lower than that of GOS-2. Therefore, the GOS-1 prepared by GOP has a low viscosity and a high relative single-layer fraction, which are beneficial to the subsequent macro-assembly of GO.

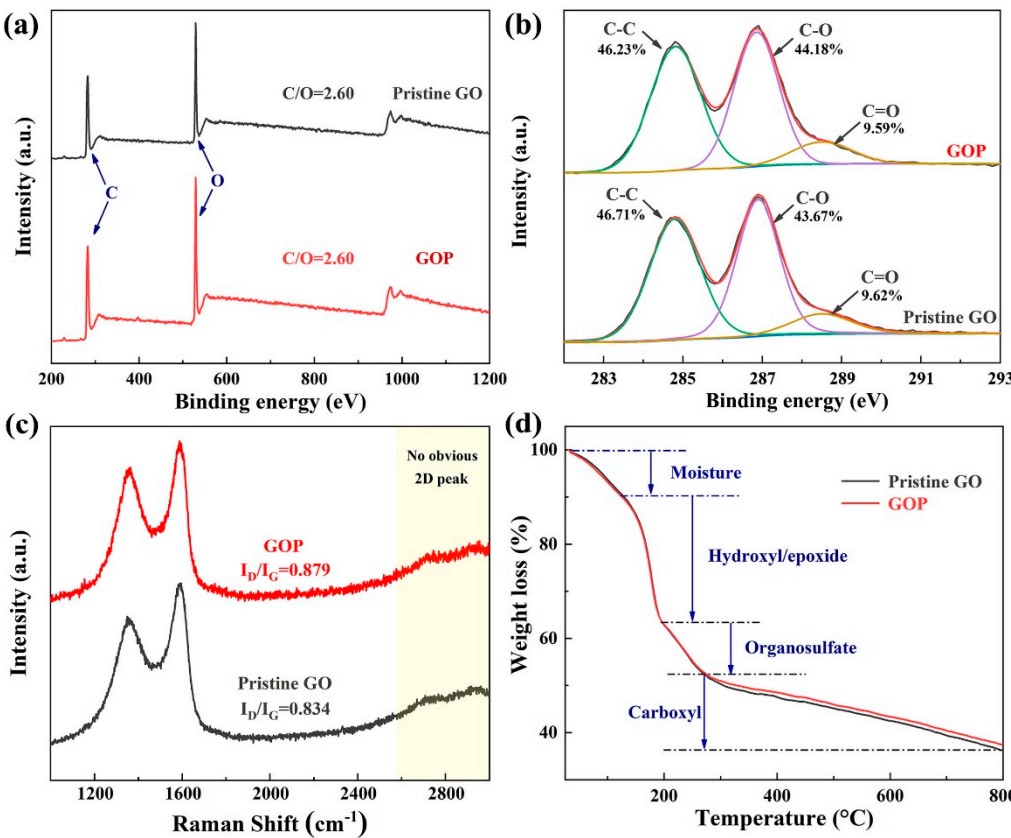

**Figure 7.** (**a**) Broad scan XPS spectra of the GOP and pristine GO. (**b**) XPS spectra for the C1s regions of GOP and pristine GO. (**c**) Reman spectra of the pristine GO and GOP. (**d**) TG curves of the GOP and pristine GO.

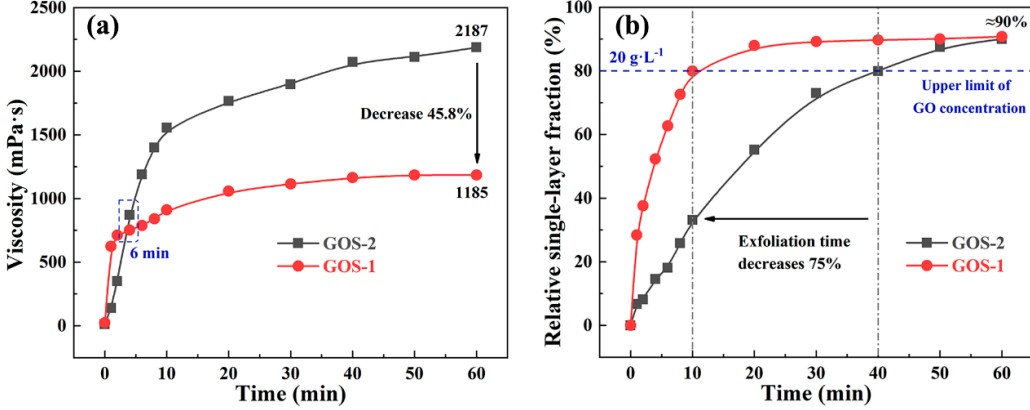

**Figure 8.** (**a**) Viscosity and (**b**) relative single-layer fraction of GOS-1 and GOS-2 with the extension of shear time.

## 4. Conclusions

In this work, we have demonstrated the optimization of the spray drying process with the RSM-BBD model for preparing high concentration well-dispersed GOS. The influences of the operating parameters, including the nozzle airflow rate, atomization pressure, and liquid feed rate on the powder yield, particle size, re-dispersibility, and adsorption capacity were studied systematically. Under the optimized condition of nozzle airflow rate 439 L·h$^{-1}$, atomization pressure 0.59 MPa, and liquid feed rate 9.0 mL·min$^{-1}$, a spray-dried GOP with powder yield 70.45%, particle size 11.65 μm, re-dispersibility 97.95%, and adsorption capacity 244.1 mg·g$^{-1}$ can be obtained. The predicted responses from RSM–BBD were validated by the experimental values. Compared with the pristine GO, the spray-dried GOP retained the original chemical properties. The GOS prepared from the GOP possessed a 45.8% decline in viscosity, a 75% reduction in shear time, and a high relative single-layer fraction of up to 90.8%. This study helps to better understand the cross-influence of operating parameters on the drying and separation effect, crumpling degree, and the physicochemical property of spray-dried GOP and sheds light on the industrial production of GOS with the "drying-redissolution" method.

**Supplementary Materials:** The following are available online at https://www.mdpi.com/article/10.3390/pr9071116/s1. Discussions about the analysis of the GOP and the calculation of the droplet size ($D_d$); Tables of the single-factor experimental results (Tables S1–S3). UV-visible spectra of GO dispersion and fitted standard GO curve (Figure S1); Effects of QL on the outlet temperature and the moisture content of GOP (Figure S2); Particle size distributions of GOPs at all the experimental conditions (Figure S3); SEM images of exfoliated GOP, pristine GO, and graphite oxide for size statistics (Figure S4); XPS spectra for the C1s regions of GOP and pristine GO (Figure S5).

**Author Contributions:** Conceptualization, L.S. and N.B.; methodology, L.S., N.B. and X.Y.; software, Y.S. and P.S.; validation, X.Y. and Y.S.; formal analysis, X.Y. and Y.S.; investigation, X.Y. and Y.S.; resources, X.Y. and Y.S.; data curation, P.S., X.Y. and Y.S.; writing—original draft preparation, X.Y. and Y.S.; writing—review and editing, L.S. and N.B.; visualization, L.S. and N.B.; supervision, N.B.; project administration, N.B.; funding acquisition, L.S. and N.B. All authors have read and agreed to the published version of the manuscript.

**Funding:** This research was supported by the National Key Research and Development Program of China (Grant No. 2020YFE0100100), Jiangsu Provincial Key Research and Development Program (No. BE2018008-1), and the Project Funded by the Priority Academic Program Development of Jiangsu Higher Education Institutions (PAPD).

**Institutional Review Board Statement:** Not applicable.

**Informed Consent Statement:** Not applicable.

**Data Availability Statement:** The data presented in this study are available in the main text of the article.

**Conflicts of Interest:** The authors declare no conflict of interest.

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
