# Peer review of "Optimization of Spray-Drying Process with Response Surface Methodology (RSM) for Preparing High Quality Graphene Oxide Slurry"

_processes, doi:10.3390/pr9071116_

Round 1

Reviewer 1 Report

In this manuscript, spray drying process was optimised and RSM-BBD model was developed for the preparation of high concentration well-dispersed Graphene oxide slurry (GOS).

Binding energy, weight loss, viscosity, and relative single-layer fractions, and the operating parameters like  nozzle airflow rate, pressure, and liquid feed rate on the powder yield, particle size, re-dispersibility, and adsorption capacity were studied.
This manuscript has novelty and can be valuable for the process engineering and the device systems where graphene oxide and its composites are used.

Sharing the Raman peaks of the prepared samples could be fine and helpful to analyze the G-peak, D-band, D’-band and 2D-band including I_D/I_G value comparison of the prepared GOS samples with the pristine Graphene oxide (GO) for discussing the deformation and disorder (defect) levels.

Related to the references:
Those comments can require references:
In Line 168:  "The re-dispersibility is related to the particle size and crumpling degree of GOP."
In Line 280: "It was reported that the re-dispersibility is closely related to its internal microstructure and the crumpling degree, which can be tuned by the droplet size controlled by the operating parameters."
References between 2 and 15 are unnecessarily long. Authors do not use them in the manuscript after the first paragraph of the Introduction. I advice to select half of them which are important for the manuscript.

There are some typing errors like  in x-axis of the Figure.S3 "Diamter", or in Equation (1) "Xij Xi".

Author Response

Response to Reviewer 1 Comments

In this manuscript, spray drying process was optimized and RSM-BBD model was developed for the preparation of high concentration well-dispersed Graphene oxide slurry (GOS).

Binding energy, weight loss, viscosity, and relative single-layer fractions, and the operating parameters like nozzle airflow rate, pressure, and liquid feed rate on the powder yield, particle size, re-dispersibility, and adsorption capacity were studied.

This manuscript has novelty and can be valuable for the process engineering and the device systems where graphene oxide and its composites are used.

Point 1: Sharing the Raman peaks of the prepared samples could be fine and helpful to analyze the G-peak, D-band, D’-band, and 2D-band including ID/IG value comparison of the prepared GOS samples with the pristine Graphene oxide (GO) for discussing the deformation and disorder (defect) levels.

Response 1: We appreciate the reviewer’s comment. The Raman spectra of the prepared GOP and pristine GO have been added, and the related descriptions have been marked in red in section 3.3.1.

The related revision is on page 13 line 341:

“Figure 7c displayed the Raman spectra of the GOP and pristine GO. It was found that the ID/IG ratio of GOP (0.879) was slightly larger than that of the pristine GO (0.834), indicating the existence of more defects and disorder levels in GOP caused by the crumpling morphology [36]. Besides, no obvious 2D peak could be found at ~2700 cm-1, which indicated no significant change in oxidation degree of GOP after spray drying.”

Point 2: Those comments can require references: In Line 168:  "The re-dispersibility is related to the particle size and crumpling degree of GOP."

In Line 280: "It was reported that the re-dispersibility is closely related to its internal microstructure and the crumpling degree, which can be tuned by the droplet size controlled by the operating parameters."

Response 2: We appreciate the reviewer’s comment. We have added the related references.

The related revision is on page 5 line 168:

“The re-dispersibility is related to the particle size and crumpling degree of GOP [16].”

And the related revision is on page 11 line 279:

“It was reported that the re-dispersibility is closely related to its internal microstructure and the crumpling degree, which can be tuned by the droplet size controlled by the operating parameters [16].”

Point 3: References between 2 and 15 are unnecessarily long. Authors do not use them in the manuscript after the first paragraph of the Introduction. I advice to select half of them which are important for the manuscript.

Response 3: We appreciate the reviewer’s comment. We have selected some references which are not essential for this manuscript.

The related revision is on page 1 line 32:

These abundant oxygen functional groups are not only beneficial to subsequent modification and dispersion processes but also serve as new active sites, endowing GO broad application prospects in energy storage and conversion, composites, adsorption, and so on [2–8].

Point 4: There are some typing errors like in x-axis of the Figure S3 "Diamter", or in Equation (1) "Xij Xi".

Response 4: We appreciate the reviewer’s comment. We have corrected the wrong word "Diamter" to "Diameter" in the x-axis of Figure S3. Equation (1) is the mathematical quadratic polynomial model in terms of the coded factors. "xi, xj" in Equation (1) represents the coded level of the independent variable.

Reviewer 2 Report

This paper is extremely well written. In my view, this paper should be accepted without any revisions in its current form as it does not even have grammatical  mistakes.

Author Response

We appreciate the reviewer’s comment.